# Evaluation of differential absorption radars in the 183 GHz band for profiling water vapour in ice clouds

Alessandro Battaglia[1,2] and Pavlos Kollias[3,4]

[1]Department of Physics and Astronomy, University of Leicester, Leicester, UK
[2]National Centre for Earth Observation, UK
[3]Stony Brook University, NY, USA
[4]University of Cologne, Cologne, Germany

**Correspondence:** Alessandro Battaglia
ab474@le.ac.uk

**Abstract.** Relative humidity (RH) measurements in ice clouds are essential for determining the ice crystals growth processes and rates. A differential absorption radar (DAR) system with several frequency channels within the 183.3 GHz water vapor absorption band is proposed for measuring RH within ice clouds. Here, the performance of a DAR system is evaluated by applying a DAR simulator to A-Train observations in combination with collocated European Centre for Medium-Range Weather Forecasts (ECMWF) reanalysis. Observations from the CloudSat W-band radar and from the CALIPSO lidar are converted first into ice microphysical properties and then coupled with ECMWF temperature and relative humidity profiles in order to compute scattering properties at any frequency within the 183.3 GHz band. Self-similar Rayleigh Gans approximation is used to model the ice crystal scattering properties. The radar reflectivities are computed both for a space/air-borne and a ground-based DAR system by using appropriate radar receiver characteristics. Sets of multi-frequency synthetic observation of attenuated reflectivities are then exploited to retrieve profile of water vapour density by fitting the line shape at different levels. 10 days of A-Train observations are used to test the measurement technique performance for different combination of tones when sampling ice clouds globally. Results show that that water vapour densities can be derived at the level that can enable ice process studies (i.e. better than 3%) both from a ground-based system (at the minute temporal scale and with circa 100 m vertical resolution) and from a space-borne system (at 500 m vertical resolution and with circa 5 km integration lengths) with four tones in the upper wing of the absorption line. A ground-based DAR system to be deployed at high latitude/high altitudes is highly recommended to test the findings of this work in the field.

## 1 Introduction

Adequate understanding of the cloud and precipitation processes that contribute to Earth's water and energy cycle is required before significant progress occur in our ability to predict future climate scenarios. This calls for a paradigm shift away from the current observing system that mainly capture snapshots of "states" to the next-generation of observing systems that can observe both states and "processes" (Stephens et al., 2018).

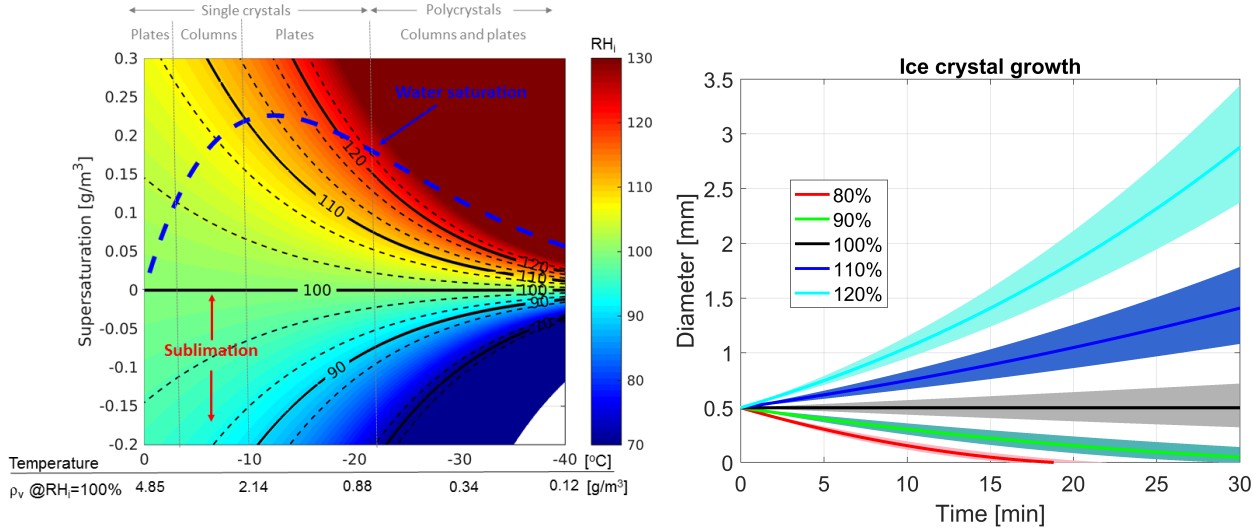

**Figure 1.** Left panel: dominant ice crystal habits (small photographs) as suggested by Bailey and Hallett (2009) for different environmental conditions as classified in terms of temperature (x-axis) and supersaturation (y-axis). The color maps the relative humidity with respect to ice, $RH_i$. The dashed blue line indicates the supersaturation of supercooled water relative to ice. Black lines correspond to different level of $RH_i$ as indicated by the labels. The dashed lines surrounding each continuous line correspond to a $\pm 3\%$ change in $RH_i$. Right panel: temporal evolution of the diameter of a 500 $\mu$m crystal environment with different supersaturation $RH_i$ (as indicated in the legend) and at T=260 K and p=500 mb. The shading corresponds to a $\pm 3\%$ perturbation in $RH_i$. The rate of mass change is assumed to be driven by diffusional growth or sublimation (description provided in Field et al. (2008)) with the Brown and Francis (1995) mass-size relationship.

Future space-borne cloud and precipitation radars are expected to be at the center of such a revolution (The Decadal Survey, 2017), thus enhancing the view depicted in the past 20 years by the TRMM Ku-band Precipitation radar (Kummerow et al., 1998), the GPM Dual-frequency (Ku-Ka) Precipitation Radar (Skofronick-Jackson et al., 2016) and the CloudSat W-band Cloud Profiling Radar (Tanelli et al., 2007). While the first Doppler radar is expected to be launched on board the EarthCARE

5    satellite in 2021 (Illingworth et al., 2015) innovative radar concepts have been studied in the past decade ranging from multi-wavelength radars proposed e.g. as payloads of the Aerosol/Cloud/Ecosystems (ACE) mission and the Polar Precipitation Measurement (PPM) mission for microphysical studies (Leinonen et al., 2015; Joe et al., 2010; Durden et al., 2016; Tanelli et al., 2018) to Doppler radars for understanding cloud dynamics (Battaglia and Kollias, 2014; Illingworth et al., 2018; Battaglia et al., 2018; Kollias et al., 2018) to constellations of radars in a CubeSat for advancing convective parameterizations (Peral et al.,

10    2015; Haddad et al., 2017; Sy et al., 2017).

In parallel, radar systems operating at much higher frequencies such as the G-band (110-300 GHz) have been proposed to study ice/snow microphysical properties (Hogan and Illingworth, 1999; Battaglia et al., 2014). Furthermore, there is interest in exploring the possibility of profiling the water vapor in cloudy areas (Lebsock et al., 2015; Millán et al., 2016; Roy et al., 2018)

by using differential absorption radar (DAR) measurements near the 183.3 GHz water vapor absorption line. Water vapor is one of the most critical atmospheric variables for numerical weather prediction models (Millán et al. (2016)) and profiles of humidity in cloudy areas are not adequately measured by current or planned systems as stated by WMO (Anderson, 2014; Nehrir et al., 2017). While Lebsock et al. (2015) theoretically investigated the possibility of profiling water vapor within the
cloudy boundary layer in presence of cumulus and stratocumulus clouds and of quantifying integrated column water vapor over ocean surfaces with a DAR system with channels on the upper wing of the 183.3 GHz absorption line, Millán et al. (2016) examined how the DAR technique can be applied to water vapor sounding in clouds at all levels by adopting multiple tones within the whole absorption band (140 to 200 GHz). A serious issue that must be considered is that international frequency allocations currently prohibit space-borne transmission at frequencies between 174.8 and 191.8 GHz due to reservation for
passive only remote sensing. Viceversa allocations are more flexible for ground-based instruments.

     Recently the DAR technique within the G-band has been demonstrated by Cooper et al. (2018): not only ground-based measurements of planetary-boundary-layer clouds have been performed but an error model and an inversion algorithm have been developed for retrieving the water vapor profile as well (Roy et al., 2018). An initial assessment of the performances of such retrieval have been performed for boundary layer clouds.

This work aims at assessing the potential of both space-borne and ground-based DAR systems with a specific focus to water vapour profiling in ice cloud studies. When coupled with that of temperature the knowledge of the water vapor density in ice clouds has two benefits.

1. It allows to derive the relative humidity with respect to ice ($RH_i$) and then to identify regions where depositional growth/sublimation processes are dominant (i.e. when the supersaturation is positive/negative in Fig. 1). Particle growth
by deposition is an important growth process in cold environments particularly when supercooled liquid water layers provide sufficient water vapor for rapid growth (i.e. in regions above the dashed blue line in the left panel of Fig. 1). The right panel of Fig. 1 shows for instance the growth rate of an initial 500 $\mu$m ice crystal for different $RH_i$ conditions clearly highlighting how the ice crystal growth rate is affected by $RH_i$.

     DAR observations could complement polarimetric radar observations like differential reflectivity that are particularly
sensitive to depositional growth in temperature regions which favor growth of asymmetric particle shapes (e.g. Verlinde et al. (2013); Oue et al. (2016)).

2. The detection and the description of supersaturation areas in high level ice clouds could help us understand how the ice crystal growth significantly enhances water mass fluxes due to sedimentation. This could have an impact on the dehydration of the air entering the lower stratosphere (Kärcher et al., 2014).

3. It may contribute to identify ice crystal habits based on the knowledge of the dominant growth in the different portions of the clouds based on thermal and moisture condition as suggested by Bailey and Hallett (2009) (dominant habits reported at the top of Fig. 1). This identification may indeed be complicated by the fact that substantial changes in habit can occur due to vertical transport caused by convection or sedimentation with ice crystals experiencing temperature changes of

tenths of K across their lifetime. Since the shape and internal mass distribution of the ice particles is affecting their scattering properties this has an immediate impact onto improving remote sensing retrievals.

The water vapor density for a given relative humidity is a strong function of temperature: for instance for $RH_i = 100\%$ the water vapor density, $\rho_v$, is changing by more than one order of magnitude (from 4.85 to 0.34 g/m$^3$, see x-axis in Fig. 1) when moving from 0 to $-30°C$. A knowledge of $RH_i$ within $5 - 7\%$ seems appropriate for identifying the relevant regimes in Fig. 1. Since $RH_i = \rho_v/\rho_{s,i}$ (with $\rho_{s,i}$ the water vapor density in saturated condition with respect to ice) the uncertainty in $RH_i$ is affected by the uncertainty in the numerator and in the denominator. The uncertainty in the denominator is driven by the uncertainty in the temperature: an error of 1 K propagates into an 8 to 10% error in $\rho_{s,i}$ with temperatures ranging from $0°C$ to $-30°C$. Uncertainties in current analyses of atmospheric temperatures are strongly regional dependent, with large uncertainties over polar, oceanic and developing nations which lack frequent radio-soundings (LANGLAND et al., 2008). Temperature uncertainties of the order of less than 1 K or better are expected from reanalysis in the middle/upper troposphere and in regions where radiosonde observations are plentiful. Advances in hyperspectral microwave sounders promise to reduce errors in temperature profiling to 0.5 K (Blackwell et al., 2011; Aires et al., 2019) and this figure is certainly at reach for ground location hosting a remote sensing observatory. This highlights that, in order to retrieve useful information for ice cloud studies, water vapor densities must be retrieved within $\sim 3 - 5\%$ or better -this in order to account for the previously mentioned additional uncertainty due to temperature- for a range of values between 0.5 and 5 g/m$^3$.

The previous study by Millán et al. (2016) has clearly demonstrated that, when dealing with DAR profiling capabilities, the main roadblock for the use of space-borne DAR measurements in process studies is represented by the precision of the measurements with potential biases being generally much smaller (e.g. see their Fig. 7). The key science question which we aim to answer in this work is therefore whether or not it is possible to beat the noise of the measurement (by averaging and by including more tones) to the level at which water vapor profiling in ice clouds could help in refining our understanding of microphysical processes. Our strategy is therefore to exploit the novel retrieval model proposed in Roy et al. (2018) in assessing the precision of DAR techniques in profiling ice clouds both from a ground and a space-borne perspective. This will allow to draw some conclusions on the potential of such observations for ice studies.

The paper is structured as following: first the theory of water vapor retrieval with DAR is shortly revisited (Sect. 2). In Sect. 3 CloudSat observations are used to reconstruct realistic ice microphysics profiles that can be used as input in a forward model for simulating reflectivities profiles at any frequency in the G-band.

Conclusions and future work are presented in Sect. 5.

## 2   Theory of water vapor retrievals

Here the theory underpinning DAR, thoroughly covered in Lebsock et al. (2015); Millán et al. (2016); Roy et al. (2018), is briefly revised. The measured reflectivity from target with effective reflectivity $Z_e(r, f)$ at a given range $r$ is given by:

$$Z_{meas}(r, f) = Z_e(r, f)\, e^{-2\tau(0 \to r, f)} \tag{1}$$

where $\tau(0 \to r, f)$ is the one way optical depth from the radar to the range $r$. The exponential term accounts for the radar attenuation due to the gases and the hydrometeors with the factor two accounting for the two way path of the radar wave. Note that multiple scattering effects (Battaglia et al., 2010; Matrosov and Battaglia, 2009) will be neglected hereafter since they are minimized by the small radar footprints (less than 450 m for the space-borne configuration) and by the low single scattering albedo of the medium at frequencies in the vicinity of the absorption line. Multiple scattering could be an issue for tones located far away from the absorption centre when encountering heavily rimed particles and it could potentially be flagged by introducing a cross-polar channel to measure linear depolarization ratio (like proposed in Battaglia et al. (2007)). As shown afterwards, such conditions, which corresponds certainly to CloudSat reflectivities exceeding 10 dBZ, are anyhow challenging for the DAR retrieval. Following Roy et al. (2018) we consider the ratio of measured reflectivities at two ranges $r_1$ and $r_2 = r_1 + \Delta r$:

$$\frac{Z_{meas}(r_1, f)}{Z_{meas}(r_2, f)} = \frac{Z_e(r_1, f)}{Z_e(r_2, f)} \, e^{-2[\tau(0 \to r_1, f) - \tau(0 \to r_2, f)]} = \frac{Z_e(r_1, f)}{Z_e(r_2, f)} \, e^{2\langle k_e(f)\rangle_{\Delta r}\Delta r} \tag{2}$$

where the $\langle\rangle_{\Delta r}$ symbol corresponds to taking the mean value for ranges between $r_1$ and $r_2$ so that

$$\langle k_e(f)\rangle_{\Delta r} \equiv \frac{\tau(0 \to r_2, f) - \tau(0 \to r_1, f)}{\Delta r} = \frac{\int_{r_1}^{r_2} k_e(r, f) dr}{\Delta r} = \frac{\int_{r_1}^{r_2} [k_{e\,gas}(r, f) + k_{e\,hydro}(r, f)] \, dr}{\Delta r} \tag{3}$$

is the mean extinction coefficient for such ranges. This equation can be further simplified by separating the water vapour components from the other gases and introducing the water vapour absorption coefficient per unit mass, $\kappa_v$ as:

$$\langle k_e(f)\rangle_{\Delta r} = \langle \rho_v \kappa_v(f, p, T)\rangle_{\Delta r} + \langle k_{e\,dry\,air+hydro}(f)\rangle_{\Delta r} \approx \langle \rho_v\rangle_{\Delta r}\kappa_v(f, \langle p\rangle_{\Delta r}, \langle T\rangle_{\Delta r}) + \langle k_{e\,dry\,air+hydro}(f)\rangle_{\Delta r} \tag{4}$$

where in the last step we have assumed that the line shape $\kappa_v(f)$ within the $\Delta r$-layer can be approximated by its value at the mean temperature and pressure of the layer and we have conjoined the dry air and hydrometeor extinction.

If we invert Eq. (3) we can then write:

$$\langle k_e(f)\rangle_{\Delta r} = \frac{1}{2\Delta r} \log\left(\frac{Z_{meas}(r_1, f)}{Z_{meas}(r_2, f)} \frac{Z_e(r_2, f)}{Z_e(r_1, f)}\right) \tag{5}$$

and recombining Eq. (5) and Eq. (4) we finally get:

$$\gamma(f, r_1, r_2) \equiv \frac{1}{2\Delta r} \log\left(\frac{Z_{meas}(r_1, f)}{Z_{meas}(r_2, f)}\right) = \langle \rho_v\rangle_{\Delta r}\kappa_v(f, \langle p\rangle_{\Delta r}, \langle T\rangle_{\Delta r}) + \underbrace{\langle k_{e\,dry\,air+hydro}(f)\rangle_{\Delta r} - \frac{1}{2\Delta r} \log\left(\frac{Z_e(r_2, f)}{Z_e(r_1, f)}\right)}_{A+Bf}$$
$$\tag{6}$$

The DAR rationale is based on the idea that by performing measurements of the left hand side of Eq. (6) at different frequencies it will be possible to fit the terms on the right hand side. The first term is directly proportional to the water vapor density via the line shape $\kappa_v(f)$; the last two terms are related to the dry air plus hydrometeor attenuation and the effective reflectivity ratio at the two ranges (thus affected by the vertical variability). They can be assumed to vary weakly with frequency. Extinction of supercooled droplet is indeed proportional to frequency (e.g. see Lhermitte (1990)) and ice crystals behaves similarly with

a linear increase with frequency, as demonstrated in Fig. 2. Note that the ice crystal attenuation is mainly driven by scattering since the single scattering albedo for all the cases here illustrated exceeds 0.95. Also the ice crystal attenuation is strongly depending on the ice crystal type (i.e. on the scattering model) as already noticed in Battaglia et al. (2014), but this dependence can be factored out in the differential method because of its distinctness from the absorption band spectral feature. Therefore the last two terms are modelled in this study via a dependence which is linear with frequency. Since the the line shape $\kappa_v(f)$ is known at a given $T$ and $p$ then $\langle\rho_v\rangle_{\Delta r}$ can be derived by a least squares fitting procedure which fits all three terms on the right in Eq. (6) to the measured $\gamma$ terms. The procedure also allows the computation of errors for the retrieved fitted parameters and of a quality index for the fitting via the normalised $\chi^2$. Note that the quantities $\gamma(f, r_1, r_2)$ are not affected by absolute calibration, which makes the whole procedure immune to calibration errors.

If only three tones are available (or the full range of tones is less than 10 GHz) then $B$ is assumed to be equal 0 (as done in Roy et al. (2018)). When only two tones are available $\rho_v$ and its error can be directly computed from:

$$\sigma_{\langle\rho_v\rangle_{\Delta r}} = \frac{1}{2\Delta r\left[\kappa_v(f_1, \langle p\rangle_{\Delta r}, \langle T\rangle_{\Delta r}) - \kappa_v(f_2, \langle p\rangle_{\Delta r}, \langle T\rangle_{\Delta r})\right]}\sqrt{\left[\Delta Z_{f_1}(r_1)\right]^2 + \left[\Delta Z_{f_1}(r_2)\right]^2 + \left[\Delta Z_{f_2}(r_1)\right]^2 + \left[\Delta Z_{f_2}(r_2)\right]^2}$$

(7)

as derived in Roy et al. (2018), where $\Delta Z$ are the uncertainties in the reflectivity measurements at the given range and frequency. This shows that the maximum achievable absolute precision in water vapor density is fixed by the given spatial resolution (i.e. the value of $\Delta r$), by the difference in the line shape $\kappa_v$ between the two tones and by the precision of the reflectivity measurements. This has the consequence that, the latter two factors being the same, lower values of absolute humidity will be measured with lower relative precision. Averaging over longer path or time improves the precision because it increases $\Delta r$ and the number of radar pulses (thus improves the precision of $Z$ measurements), respectively. Adopting multiple tones allows to improve the precision for a range of water vapor densities by finding the right balance between large differences in $\kappa_v$ and good precision in the reflectivity signal, i.e. good signal to noise ratio (SNR). When multiple tones are involved, the line-fitting retrieval routine implemented in this work gives an estimate of the measurement precision by generalising Eq. (7).

## 3   Simulation of DAR profiles from CloudSat data

At present, no radar reflectivity measurements at multiple G-band tones are available that can be used to evaluate the performance of the technique. Our approach relies on using retrieved ice microphysical properties from spaceborne sensors and use them as input to a forward radar model (DAR model) to generate reflectivities around the 183.3 GHz absorption band.

The CloudSat 94 GHz (3.2 mm) Cloud Profiling Radar (CPR) provides global observations of ice cloud profiles at a vertical resolution of 480 m and a cross-track/along-track horizontal footprint of 1.5 km×2.5 km (Tanelli et al., 2008). When integrated with the observations from the CALIPSO Cloud-Aerosol Lidar with Orthogonal Polarization (CALIOP) (Winker et al., 2007) such observations can be used to retrieve ice microphysics. Here retrievals adopting the DARDAR algorithm (Delanoë and Hogan (2010), http://www.icare.univ-lille1.fr/projects/dardar/) are used as input for the the DAR modelling. ECMWF auxiliary data are used as input for temperature, pressure and relative humidity.

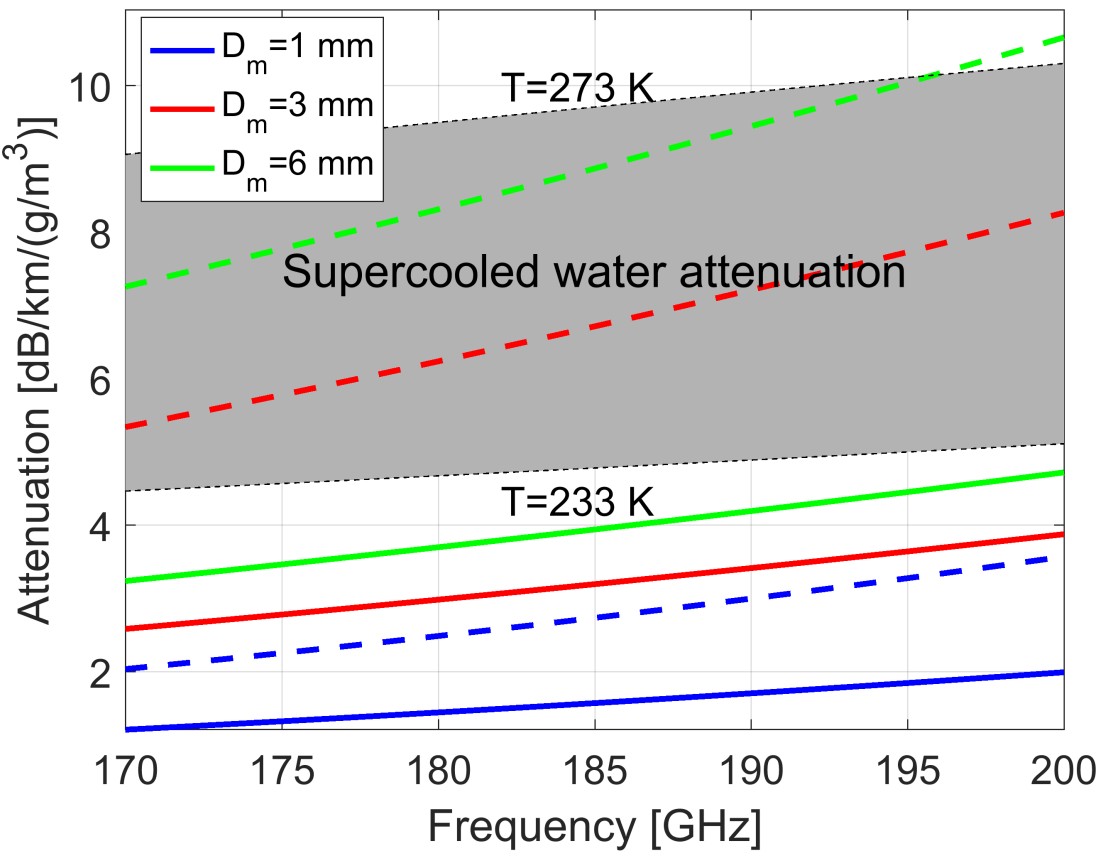

**Figure 2.** Attenuation coefficient for ice crystals with different mass-weighted maximum particle diameters as indicated in the legend for the frequency range of interest for this study. Exponential size distributions have been assumed. Dashed and continuous lines correspond to the model "A; $LWP = 0.1kg/m^2$" from Leinonen and Szyrmer (2015) and to the Hogan and Westbrook (2014) model, respectively. The grey shaded area corresponds to the attenuation coefficient for supercooled liquid clouds for temperatures in the range between $-30°C$ and $0°C$. Water refractive index is computed according to the Ellison07 model, see Turner et al. (2016).

The DAR forward model uses the millimeter-wave propagation model from Rosenkranz (1999) for gas attenuation whereas the self-similar Rayleigh-Gans scattering model (Hogan and Westbrook, 2014) is adopted for computing the scattering properties of ice particles. This approach has the clear advantage that scattering properties can be computed at any frequency with practically no computational cost. The ice crystals model proposed by Leinonen and Szyrmer (2015) and labelled as model "A; $LWP = 0.1kg/m^2$" is used to derive the parameters for the self-similar model by taking into account the internal structure of the aggregates. Tridon et al. (2019) have shown that the scattering properties generated via this methodology generally well fits triple frequency radar measurements and in situ measurements.

**Table 1.** Technical specifications of the DAR space-borne system used in this study. The configuration here adopted is the one proposed in an on-going UK-CEOI study (Dr Duncan Robertson, personal communications).

| | |
|---|---|
| Satellite altitude, $h_{sat}$ | 500 km |
| Satellite velocity, $v_{sat}$ | 7600 ms$^{-1}$ |
| Frequency | 170-200 GHz |
| Transmit power | 100 W (EIK technology) |
| Antenna diameter | $\geq 2$ m |
| Antenna beam-width, $\theta_{3dB}$ | $\leq 0.05°$ |
| Antenna gain | 70 dBi |
| Receiver Noise Figure | 6 dB |
| Pulse width | 3.3 $\mu$s |
| Pulse Repetition Frequency (with frequency diversity) | 6 kHz |
| Single pulse sensitivity | -22 dBZ |

**Table 2.** Specifics of the frequency-modulated-continuous wave radar based on W-band power amplifier and GaAs Schottky diode frequency multiplication (Nils et al. (2017)) for the ground-based simulation (Dr Peter Huggard, personal communications).

| | |
|---|---|
| Frequency | 170-200 GHz |
| Transmit power | 200 mW |
| Antenna diameter | 0.4 m |
| Antenna beam-width, $\theta_{3dB}$ | $\leq 0.3°$ |
| Antenna gain | 55 dBi |
| Receiver Noise Figure | 6.5 dB |
| Chirp Repetition Frequency | 6 kHz |
| Bandwidth | 2 MHz |
| Range resolution | 75 m |
| Minimum detectable reflectivity @1km range and 1 s integration | -50 dBZ |

Noise is injected into the reflectivity measurements according to the formula (see Appendix in Hogan et al. (2005)):

$$\Delta Z[dB] = \frac{4.343}{\sqrt{N_p}} \left[ max\left(1, \frac{\lambda}{4\sqrt{\pi}\sigma_v\tau_s}\right) + \frac{2}{SNR} + \frac{1}{SNR^2} \right]^{1/2} \tag{8}$$

where $N_p$ is the number of transmitted radar pulses (e.g. in the space-borne configuration 4200 for an integration length of 5 km), $\tau_s$ is the time between samples (i.e. the reciprocal of the pulse repetition frequency) and $\sigma_v$ is the spectral width of the Doppler spectrum. For space-borne systems the first term inside the bracket is practically always close to one because the Doppler spectral width is expected to exceed 2 m/s due to the large satellite velocity (see Eq. 6 in Battaglia and Kollias (2014)). The first term inside the square bracket needs to be at least one because the number of independent samples has to be

smaller or equal to the number of samples. This implies that the so-called "time to independence" is of the order of 100 $\mu$s, thus smaller than the time between pulses (equal to 166.7 $\mu$s for a PRF=6 kHz). The single pulse sensitivity is assumed to be $-22$ dBZ, a realistic value with current technology (see Tab. 1). For ground-based system on the other hand we have assumed a spectral width equal to 1 m/s and a single pulse sensitivity of -50 dBZ at 1 km range with 1 s integration (see Tab. 2). The DAR system shown in Tab. 1 is for a 2% duty cycle, 100 W peak output power Extended Interaction Klystron (EIK) system. The Communications and Power Industries (CPI) has about 15 GHz bandwidth. Selecting a number of tones (e.g. four) is technologically feasible using a single chirp generator and four intermediate chains which are switched between to select the tone. The switching can be done from pulse to pulse.

In addition to high power sources (EIK), lower power sources are available either using frequency multipliers coupled with commercially available amplifiers or microwave sources/oscillators (Virginal Diodes, https://www.vadiodes.com/). Recently, Jet Propulsion Laboratory developed a new power approach using GaAs Schottky diode frequency two-time multipliers at 183 GHz (Cooper et al., 2016). Preliminary estimates of the expected radar sensitivity using these different architectures indicate that a minimum sensitivity of -22 dBZ is possible for four different tones. The impact of reducing the number of samples to accommodate additional tones is discussed in Sect. 4.

### 3.1 Case study

The methodology is demonstrated for a precipitating system observed by CloudSat over the Southern Ocean between Antarctica and South America on the 2nd January 2007 at about 20:16 UTC. The system extends for roughly 1300 km with temperature at the surface ranging from 281 K at the southern edge of the system to 274 K at the the northern edge of the system. The CloudSat 94 GHz reflectivity as derived from the 2B-GEOPROF product (Mace et al., 2007) is shown on the top left panel of Fig. 3. The zero isotherm clearly demarcates the ice vs the liquid transition. The co-located ECMWF reanalysis for the relative humidity field with respect to ice is depicted in the top right panel. In the glaciated region of the precipitating system the synergy between the CloudSat radar and the CALIPSO lidar (Sassen et al., 2008) offers a unique prospective on the ice microphysics (Battaglia and Delanöe, 2013). The outputs of the DARDAR retrieval (Delanöe and Hogan, 2010) are shown in the bottom panels of Fig. 3.

These microphysical outputs are then used with look-up-tables generated from scattering models to compute reflectivities at any frequency within the 183.3 GHz absorption line. Examples of two pair of frequencies (187 and 200 GHz for the space-borne and 186.3 and 200 GHz for the ground-based configurations, respectively) are shown in Fig. 4. It is interesting to note how differently the two frequencies are penetrating into the precipitating system, with the 187 GHz (186.3 GHz) severely attenuated by water vapour below 4 km (above 3 km) in the space-borne (ground-based) configuration. On the other hand the 200 GHz is clearly attenuated in the region below 2 km at latitudes between -60° and $-58°$, a combined result of large ice water and water vapor contents.

The profile at latitude -58.07° (black arrow in the top left panel of Fig. 3) is used here to demonstrate how to derive a water vapor profile in a three-step procedure (see Fig. 5):

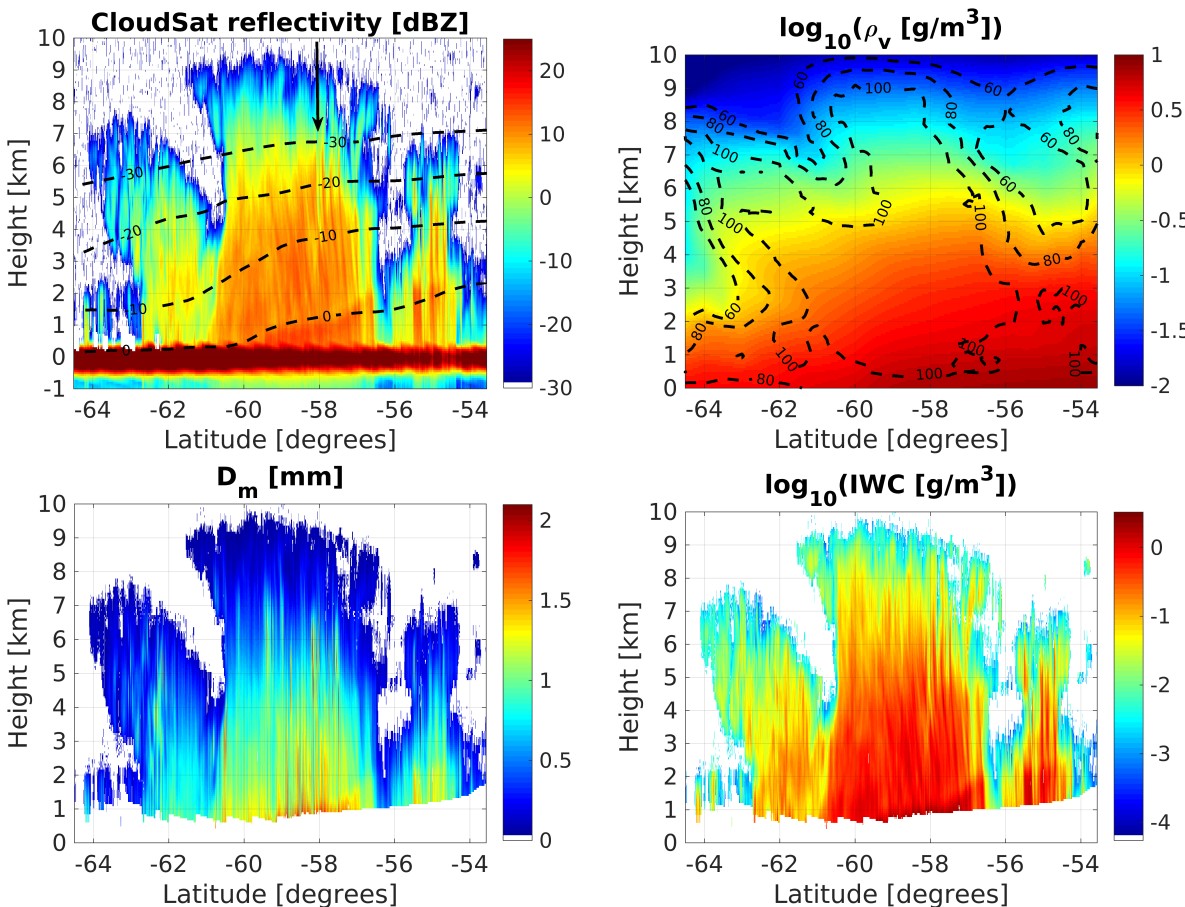

**Figure 3.** Top left: CloudSat measured reflectivity in the Southern Ocean south-west of Cape Horn. Dashed black lines corresponds to different isotherms as labeled while the black arrow corresponds to the profile analysed in Fig. 5. Top right: water vapor density as derived from ECMWF reanalysis with regions of constant relative humidity with respect to ice depicted as dashed lines. Bottom panels: mean mass-weighted diameter of ice particle (left) and ice water content (right) as retrieved by the DARDAR product.

1. an interval $\Delta r$ is selected and the profiles of the quantity $\gamma(f_j, r)$ [see Eq. (6)] are computed with their corresponding errors [computed from the estimated errors on the measured reflectivities via Eq. (8)] at the different DAR frequencies $f_1, f_2, \ldots$ (continuous blue lines with bars in the small insets of Fig. 5);

2. the spectral dependence of the line shape $\kappa_v(f, \langle p \rangle_{\Delta r}, \langle T \rangle_{\Delta r})$ is derived at each level (dashed red lines in the small insets of Fig. 5) by using the average temperature and pressure of the layer and the gas absorption model;

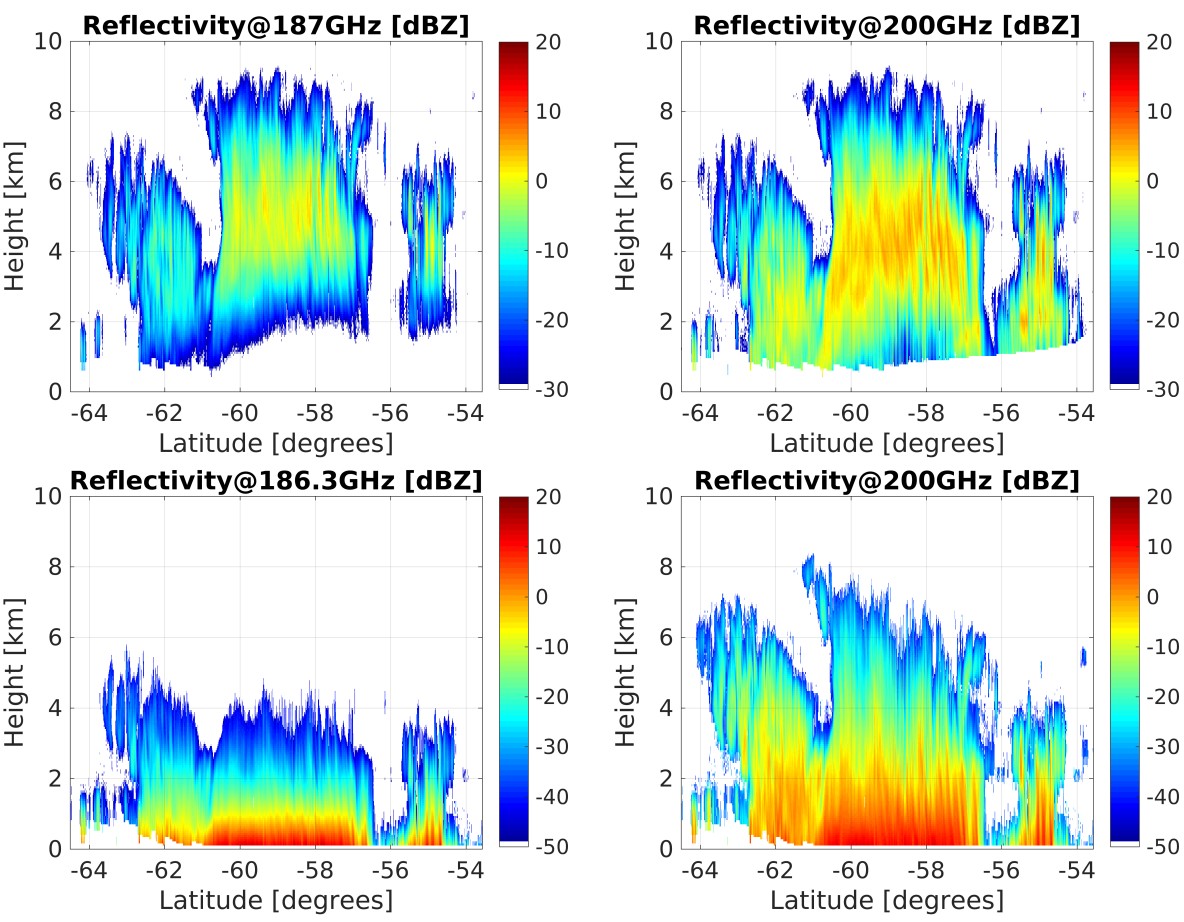

**Figure 4.** Top (bottom) panels: simulated reflectivities at 187 and 200 GHz (186.3 and 200 GHz) for a space-borne (ground-based) system with specifics as in Tab. 1 for the scene shown in Fig. 3. The ground-based system is assumed to be located at the 270 K isotherm line drawn in the top left panel of Fig. 3. Note the different ranges in the reflectivity colorbars of top and bottom panels driven by the better sensitivity achieved by the ground-based system at short ranges.

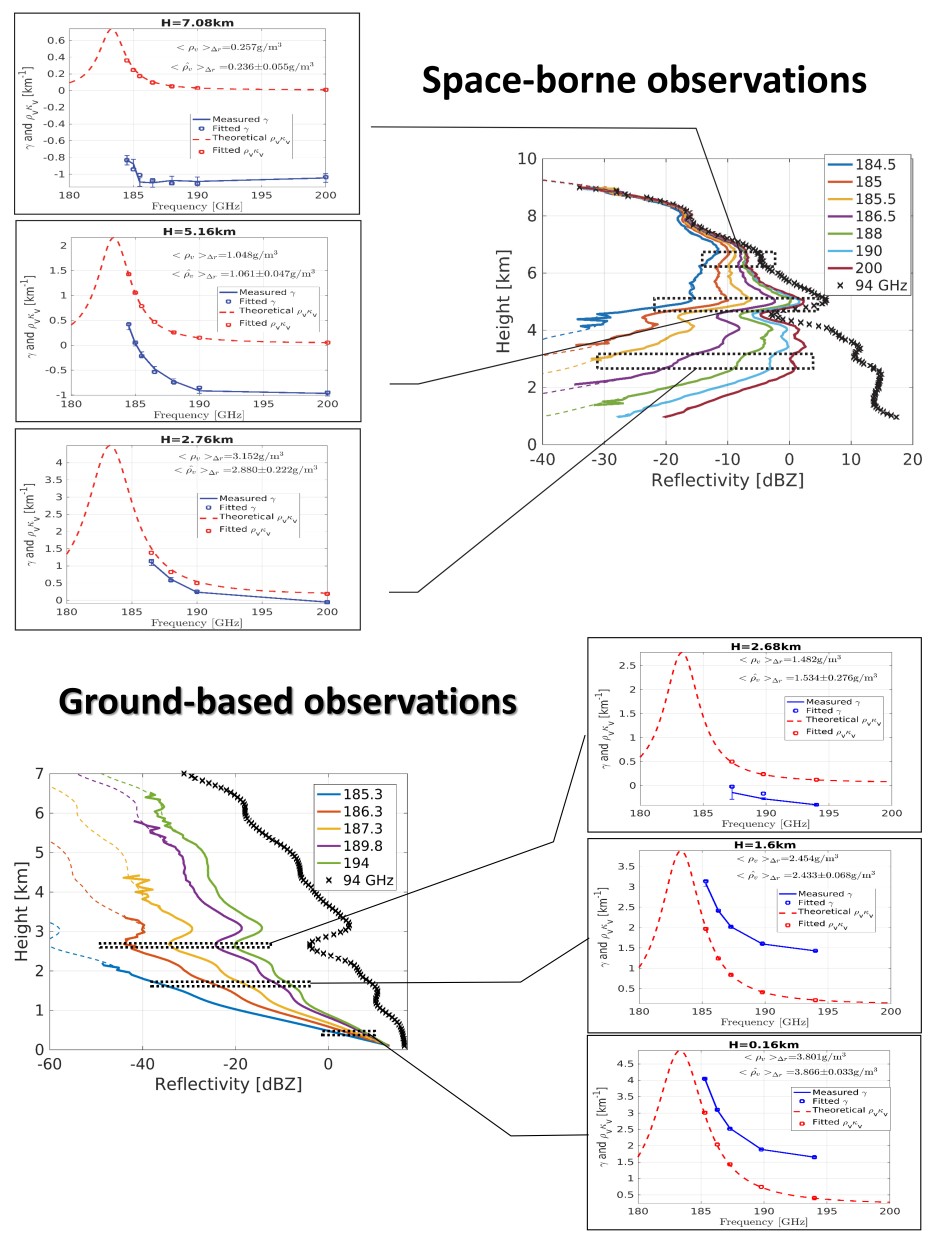

**Figure 5.** Top panel: simulated reflectivities for the profile at latitude -58.07° (black arrow in the top left panel of Fig. 3) for a 7-channel space-borne DAR with frequencies on the upper wing of the 183.3 GHz line. An integration length of 1.1 km is assumed (corresponding to $N_p = 920$). The CloudSat 94 GHz profiles is shown for reference as well (black crosses). Continuous (dashed) lines correspond to reflectivities including (without) noise. Three panels: examples of the fitting procedure at three different altitudes to estimate $\langle \rho_v \rangle_{\Delta r}$ with $\Delta r = 500\,m$. True and estimated values are inserted in the figure. Bottom panel: same as top panel side for a 5-tone ground-based DAR. An integration time of 2 min (corresponding to $N_p = 720,000$) and a vertical resolution of 120 m are assumed.

3. a least squares fitting procedure of the form expressed in Eq. (6) which accounts for the errors in $\gamma(f_j, r)$ allows to retrieve estimates of the three fitting parameters ($\hat{A}, \hat{B}$ and $\langle \hat{\rho}_v \rangle_{\Delta r}$). $\gamma$ values that are too noisy are excluded from the fitting (e.g. at 2.76 km only four tones are considered for the space-borne configuration).

For the space-borne configuration the retrieval shows that a set of 7-tone DAR with frequencies on the upper wing of the 183.3 GHz band as listed in the legend of the top panel of Fig. 5 can retrieve water vapor within the ice cloud with good precision (i.e. within $15\%$) between 7.0 km (240 K) down to 2.5 km (268 K) with water vapour contents changing by more than one order of magnitude. The relative error in the retrieval of $\rho_v$ for the whole case study shown in Fig. 3 is reproduced in the top panel of Fig. 6. Clearly there are two critical regions: 1) at low temperatures ($\approx T < -30°$C) low values of $\rho_v$ limit the amplitude of the signal (e.g. compare the red curves between the top three small insets in Fig. 5); 2) at warm temperatures ($\approx T > -10°$C) and large CloudSat reflectivities the cumulated attenuation tends to strongly reduce the SNR and therefore [see formula (8)] increase the uncertainty of the reflectivity measurements and as a result of $\gamma(f_j, r)$. In both situations the retrieval errors become large but such deterioration can be clearly identified by looking at the SNR of the different DAR channels and at the associated error induced in the estimated value of water vapor, $\langle \hat{\rho}_v \rangle_{\Delta r}$.

The same profile has also been used to analyze the performance of a ground-based instrument by assuming that the instrument is located at the $-3°C$ isothermal line and is looking upward. Again tones in the upper wing of the absorption band are selected. The simulated reflectivities, shown in the bottom panels of Fig. 4-5, show strong attenuation in the lower troposphere with the tones close to the center of the line reaching the noise level already just above 2 km. The only tones that can penetrate deep into the clouds are the ones that have not enough water vapor signal high up in the troposphere (e.g. the highest three tones at 2.68 km, see bottom small insets in Fig. 5). This demonstrates why, while the precision of the retrieval in the lower troposphere is excellent, it deteriorates quickly above 2.5 km. The bottom panel of Fig. 6 demonstrates the same thing for the whole event: the precision of the retrieval is quickly worsening 2/2.5 km above the ground where temperatures decrease to values lower than -15$°C$. On the other hand, by integrating for periods of the order of 1-2 minutes, ground-based system can achieve extremely accurate results for temperature between 0 and -15$°C$.

Compared to the space-borne set-up things are expected to substantially improve when dealing with an airborne configuration (bottom panels). The key advantages are: 1) better sensitivity because of closer distance to the target; 2) slower platform speed that allows to collect more pulses for the same integration length. As a result the precision and/or the resolution of the retrieval are significantly improved compared to the space-borne configuration (contrast the top left and bottom panels). The two bottom panels demonstrate the trade off between long and short pulses for the air-borne mode. In the left panel a pulse which is four times shorter than that in the right panel is adopted (i.e. 120 vs 480 m). As a result the sensitivity is 12 dB (a factor of 16) better in the latter case, which translates in a much better precision of the retrieval. Thus it is recommended to use pulses that match the required vertical resolution because SNR is a critical parameter for the precision of the measurement. Averaging more gates do not recover the same precision.

This case study highlights that sounding ice clouds by air-borne or space-borne DAR systems is clearly advantageous with respect to ground systems because regions with low water vapor contents (thus low attenuation) are encountered first. This implies that tones close to the line center can stay well above MDT in the areas where they provide useful information (i.e. at

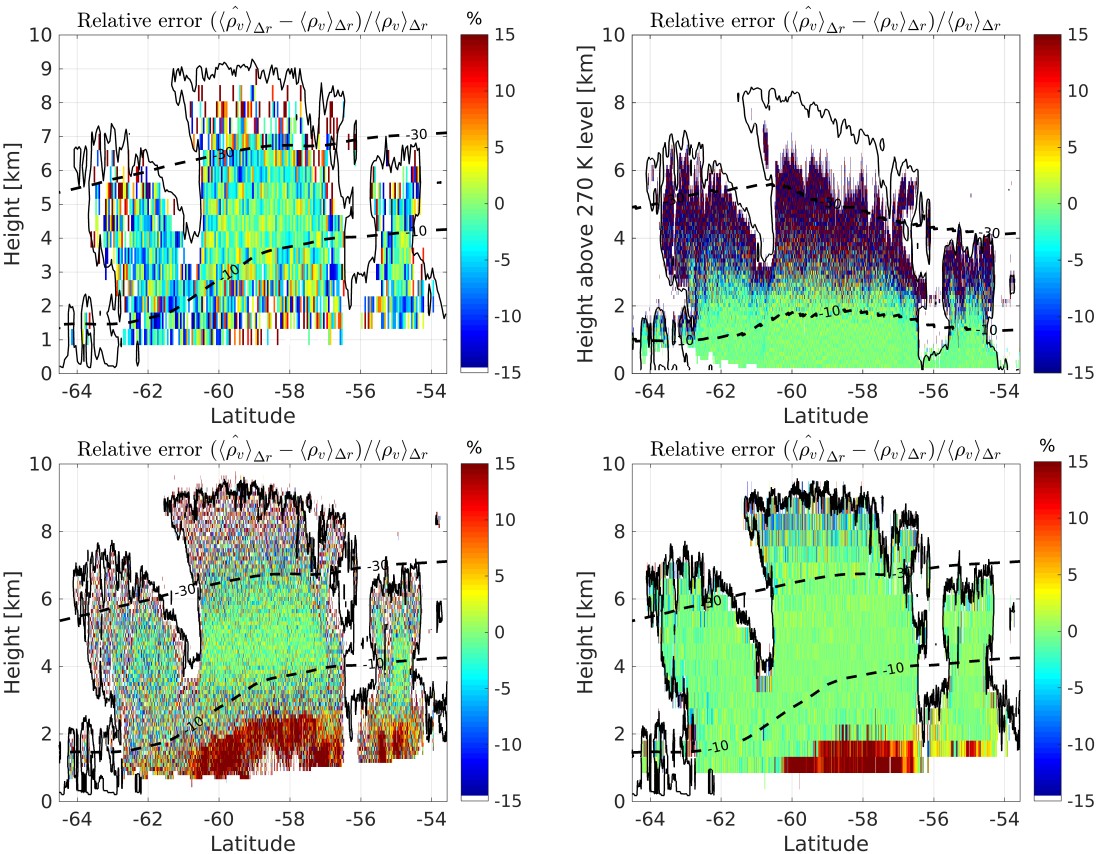

**Figure 6.** Top left panel: relative error in the retrieval of $\rho_v$ for the case study shown in Fig. 3 for a 7-channel space-borne DAR with frequencies as listed in the legend on the top side of Fig. 5. Here $\Delta r = 480$ m and a 5 km along-track averaging has been performed. The dashed lines correspond to the -30°C and -10°C isotherms and the black line corresponds to CloudSat reflectivities of -25 dBZ (roughly indicating the cloud boundaries). Top right panel: same as left panel for a 5-tone ground-based DAR with frequencies as listed in the legend on the bottom side of Fig. 5. Here $\Delta r = 120$ m and a 2-minute averaging has been performed. Bottom panels: same as top left panel for an airborne system with $\Delta r = 120$ m (left) and $\Delta r = 480$ m (right) and a 1 km along-track averaging. The single pulse sensitivity is assumed to be -33 dBZ (left) and -45 dBZ (right) at 1 km distance.

low water vapor contents). The same is not true for ground-based geometry because, unless the temperature at the ground is very cold, large levels of attenuation are experienced by the radar pulse in the lower troposphere.

## 4   Statistical analysis from CloudSat climatology

The A-Train has provided the first global climatology of ice clouds with a detailed description of ice cloud occurrences, ice microphysics and ice radiative effects (Hong and Liu, 2015). The A-Train ice cloud dataset represents therefore an ideal test-bed to investigate the potential of a DAR system for measuring relative humidity inside ice clouds. The methodology described

in Sect. 3 has been applied to ten days of CloudSat data (from 1st to 10th January 2007) to study the performances both of a

space-borne and a ground-based DAR system with several channels within the 183.3 GHz absorption band. The ground-based system is assumed to look upward from the height corresponding to the $270\ K$ isothermal level, as identified by the ECMWF reanalysis. For any profile with ice water path exceeding 20 g/m$^2$ the profile of water vapour is retrieved via the DAR technique and, by comparing such value with the assumed one (from ECMWF reanalysis), the relative error on $\rho_v$ is computed. Results are binned according to the CloudSat reflectivity values (above -10 dBZ and -25 dBZ for the space borne and ground-based

system, respectively) and the ambient temperatures (above 240 K). Fig. 7 shows the fractional occurrence when the DAR systems provide $\rho_v$ with precision better than $3\%$ (i.e. a very valuable information). For the space-borne system there is an optimal region between -5 and 15 dBZ and for temperatures between 250 and 265 K. Results tends to worsen at temperature close to 273 K and at very high W-band CloudSat reflectivities, which occur typically at higher temperatures (a result of the reduced number of tones with signal significantly above the noise floor), but also at very cold temperature (a result of the

reduced absorption for tones further away from the band center) and low CloudSat reflectivities (a result of the reduced SNR).

For the ground-based system (right panels in Fig. 7) $\rho_v$ is optimally retrieved in the lower troposphere with the quality of the retrieval typically worsening with decreasing temperatures and decreasing reflectivities (due to the reduced SNRs). The only exception is at very large reflectivities, where non linearities of the right hand term in Eq. (6) introduced by Mie and attenuation effects cause larger errors.

We have selected different combinations with 2, 3, . . . 5 tones and we have analysed which combinations achieve the best retrieval performances. As a first step we have assumed that the sensitivity of the system does not change when increasing the number of tones. This is the case if the duty cycle of the radar system could be increased accordingly and frequency diversity could be implemented. Otherwise the sensitivity of each channel is going down with $\sqrt{N_{tones}}$ becasue of the reduction in the number of samples $N_p$, and the effect will be discussed later. Results are summarized in Fig. 8. Clearly increasing the number

of tones (all with the same sensitivities) is beneficial but the improvement when surpassing four tones is marginal (e.g. compare the 4 with the 5 and 6 tones). On the other hand it is obvious that improving the SNR is generally producing better results via a reduction of the noise in the reflectivity measurements according to Eq.( 8). For instance for the two- and four-tone curves the impact of the improvement corresponding to a variation of a factor of two in sensitivity ($\pm 3$ dB) is illustrated in Fig. 8 by the shading. As a result there is indeed an improvement in water vapor profiling when using four vs two channels. In fact there is the obvious advantage that with four tones it is possible to perform the three-parameter fit of Eq. (6), thus avoiding

5   the biases introduced by frequency-dependent hydrometeor scattering effects. This remains true even when considering DAR configurations with the same duty cycle. In that case, doubling the number of channel corresponds to averaging half the number of samples, which equates to a reduction of 1.5 dB in sensitivity (so roughly half the range currently shown by the shaded area). But the blue shaded region remains well above the red shaded one.

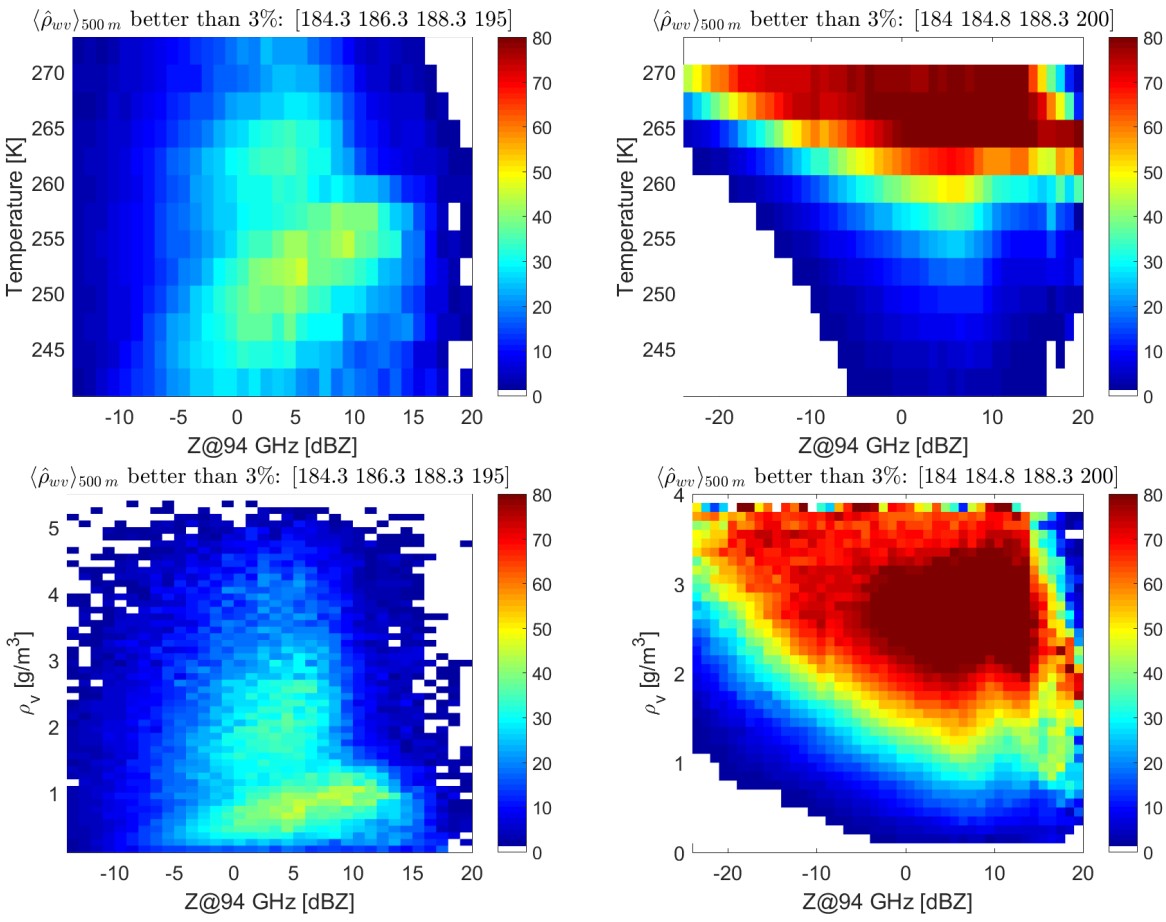

**Figure 7.** Statistical analysis based on 10 days of CloudSat showing the expected frequency occurrence of retrievals of $\rho_v$ better than $3\%$ for a space-borne system (left) and a ground-based system (with ground temperature of 270 K). Top (bottom) panels: results are clustered using reflectivities vs temperatures (water vapor contents). The specifications of the systems correspond to 4-tone DARs which are optimized for ice cloud studies.

## 5 Discussion and conclusions

The potential of a multi-frequency differential absorption radar (DAR) system with several tones within the 183.3 GHz water
5  vapor absorption band for profiling water vapour within ice clouds is assessed both for a ground-based and a space-borne configuration. Realistic ice profiles derived from A-Train observations are inputs of DAR simulations which are used to test the precision performances of water vapor retrievals based on fitting the line shape via a minimum least square fitting procedure.

Our findings can be summarized as following.

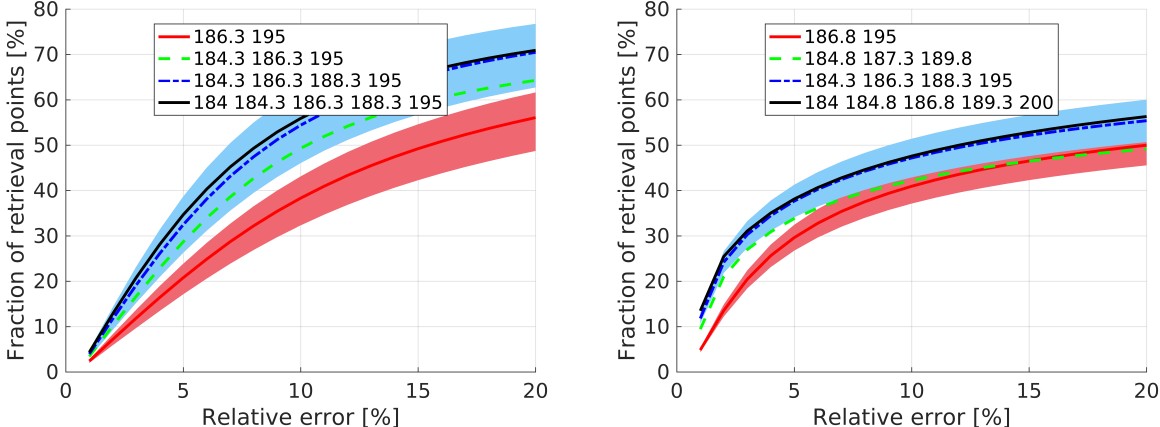

**Figure 8.** Fraction of retrieval points (y-axis) having errors lower than a given threshold (x-axis) for the space-borne configuration (left) and the ground-based configuration with 2-3-4 and 5 tones. Only the combinations that achieve the best accuracies (as indicated in the legend) are reported. The shaded region indicates results when the sensitivity is increased/decreased by 3 dB. For the space-borne (ground-based) configuration the retrieval is applied only to points corresponding to CloudSat reflectivities exceeding -15 dBZ (-25 dBZ) and temperature exceeding 240 K.

1. With realistic minimum detection thresholds, DARs can provide useful information in thick ice/mixed-phase clouds and they can complement other techniques (e.g. water vapor DIALs, Nehrir et al. (2017)). Four tone DARs seem to be the right balance between complexity (i.e. number of channels) and retrieval performances. In the domain of CloudSat reflectivities above -15 dBZ and $T > 240$ some of the best 4-tone combination allow to retrieve $\rho_v$ with precision better than $3\%$ in more than $25\%$ of the cases when ice is present with the best results obtained for ice clouds with reflectivities between -5 and 10 dBZ.

2. Ground-based DAR systems can provide excellent profiling of the warmer parts of ice clouds where $\rho_v$ values exceed $1 \mathrm{~gm}^{-3}$ but they become increasingly less precise when looking at the cold regions with low moisture. In such areas things are expected to improve when colder ground temperature are considered. In this study we have simulated a scenario with ground temperature of 270 K with global climatology. Of course the selection of the tones could be optimized for a specific location and time of the year based on the local cloud and temperature climatology. Also scanning options could be considered to increase the differential absorption signal of channels far away from the center of the band by increasing the path length.

3. Air-borne or space-borne DAR systems are clearly advantageous with respect to ground systems when looking at regions with low water vapor contents because such regions are encountered first by the radar wave and therefore are affected by less attenuation. This implies that tones close to the band center can stay well above MDT in the areas where they

provide useful information (i.e. at low water vapor contents). The same is not true for ground-based geometry because, unless the temperature at the ground is very cold, large levels of attenuation are experienced by the radar tones close to the band center in the lower troposphere.

4. Because SNR is a critical parameter for the precision of the measurement the selection of the radar resolution should ideally match the vertical resolution required for the water vapor product.

5. The selection of the tones is driven by a tradeoff between differential signal and signal. Ideally the attenuation signal should be maximised but if the attenuation is too strong the signal becomes increasingly noisy and ultimately goes below the minimum sensitivity. For ground-based systems it would be ideal to have tones that can be adjusted depending on the atmospheric conditions and latitude/altitude location since, with lower ground temperatures, channels closer to the 183.3 GHz center becomes increasingly useful.

6. The quality of the retrieval can be easily evaluated by considering retrieval errors and $\chi^2$ values that are computed as part of the minimum least square fitting procedure.

7. Transmitting licences are attainable for airborne and ground-based (e.g. in UK DAR tones within the following bands may be allowed: 173.85 to 182 GHz, 185 to 190 GHz, 191.8 to 195.75 GHz, 196.15 to 199.99 GHz with other allowed windows below 173.85) but currently much more unaccessible for space-borne systems since such bands are reserved to passive microwave radiometers. As a first step to assess the potential of the DAR concept for ice cloud studies and to properly evaluate its accuracy (via comparison with radio-soundings) it is highly recommended to deploy a ground-based DAR system at high latitude/ high altitudes.

*Acknowledgements.* This work was supported by the European Space Agency under the activity "Multi Frequency Radar Instrument Study", Contract: 4000120689/17/NL/IA. The work by Alessandro Battaglia has been supported by the project Radiation and Rainfall funded by the UK National Center for Earth Observation. This research used the ALICE High Performance Computing Facility at the University of Leicester.

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
