# Peer review of "Evaluation of differential absorption radars in the 183 GHz band for profiling water vapour in ice clouds"

_Atmospheric Measurement Techniques, 2019_

## Referee Comment (RC1) · Richard Roy (Referee) · 12 Mar 2019

The manuscript builds on previous theoretical and experimental studies of differential absorption radar near the 183 GHz water vapor absorption line, with a central focus on the retrieval of water vapor density inside of ice clouds. Specifically, it extends the multi-frequency line-fitting retrieval method from Roy et al. 2018 to include a linear term in the measured differential absorption coefficient in order to account for frequency-dependent scattering from hydrometeors. This retrieval is implemented in the context of spaceborne and ground-based instrument simulators that utilize CloudSat micro-physical products in conjunction with ECMWF reanalysis fields (as done in Millan et

al. 2016). The ice crystal scattering calculations employ a more sophisticated method (the self-similar Rayleigh-Gans approximation) than that used in previous DAR simulation studies (Mie), and thus provide a more realistic picture of what a DAR would measure in the presence of ice clouds. Furthermore, this study explores radar transmit frequencies on the high-frequency side of the water line peak, while previous studies have focused on the low-frequency side. The results are presented and discussed in a thorough manner, and the manuscript is generally well organized and written. In terms of scientific impact, measurements from such a proposed spaceborne DAR instrument would provide important observations for ice cloud studies.

I have a few comments/questions that I would like the authors to address.

1. As a general comment for G-band radars, it is important to acknowledge the international frequency allocation restrictions that prohibit transmission within certain frequency ranges. Since this has impacted similar technologies in recent years, it seems reasonable to address this issue in the introduction to the paper.

2. Page 4, Line 1: The discussion of relative humidity measurement precision only addresses the role of absolute humidity measurement uncertainty. However, the temperature uncertainty is likely to dominate the error in RH. For instance, assuming the absolute humidity (i.e. water vapor density) is known perfectly, an error of 1 K in the temperature would lead to an error of 8% in RH at 260 K. This is much larger than the 3% level that is used as a metric for good accuracy in the paper. How would coincident temperature measurements be performed for precise RH studies? Since a principal goal of the paper is to show the utility of DAR for retrieving RH (and not simply water vapor density) for ice microphysics studies, this point needs to be addressed.

3. Page 5, Line 9: The authors use the phrase "mean square fitting procedure" a couple times in the text. Do they mean to say "least squares fitting procedure"?

4. Page 5, Line 11: What sets the necessary frequency span to be 10 GHz if one wants to allow for the linear term with coefficient $B$? Shouldn't this depend on where the window of frequencies is positioned relative to the line center?

5. Page 7, Line 1: The issue of sacrificing duty-cycle for an increased number of frequencies in DAR is very important. It is unclear what the authors mean when saying that sensitivity can be held constant while increasing the number of frequencies by "increasing the duty cycle of the radar." Presumably, one wouldn't want to sacrifice the current range resolution of roughly 500 m, which means the pulse width cannot be lengthened from 3.3 $\mu$s. Additionally, since the pulse time-of-flight through 10 km of atmosphere takes roughly 70 $\mu$s (2-way), how can the PRF be increased much from 6 kHz?

Furthermore, the implementation of frequency diversity is technologically non-trivial for the large number (up to 7) and range ($\approx 15$ GHz) of frequencies proposed in this work. Since the large frequency range is critical for the 3-parameter fitting routine, and since a reduction in the sensitivity per channel by a factor of $\sqrt{7}$ would certainly affect the DAR measurement precision, a comment on the technical feasibility of such a system is needed.

6. Page 9, Line 15: The statement that water vapor can be retrieved to better than 15% accuracy in regions of the atmosphere where the water vapor density varies by more than one order of magnitude can be misleading. An important property of the DAR measurement method is that the measurement parameters (specifically the transmit frequency locations and number of pulses per frequency) and the local pressure and temperature determine the maximum achievable absolute precision in water vapor density for a given spatial resolution (i.e. value of $\Delta r$). This has the consequence that for a given measuring system, lower values of absolute humidity will be measured with lower relative precision. The important relationship between absolute humidity value and relative precision of the DAR measurement should be discussed.

As a related point, while the line-fitting retrieval routine implemented in this work gives an estimate of the measurement precision, this information is not discussed. The authors do go into great detail about the measurement biases (i.e. accuracy) in the form of relative error plots, but do not present results on the measurement precision. It would be useful to include a corresponding plot of measurement precision in Fig. 6 along with the relative error plot. Do the statistical errors from the line-fitting procedure agree well with the scatter of measured values relative to "truth." Such a discussion would elucidate the difference between systematic biases and random error in the measurement.

7. Page 12, Line 19: It is confusing to read that results worsen for high reflectivities, where one usually expects the SNR to be large. Maybe what is meant here is that regions of high reflectivity are also associated with high absorption (or are lower in the atmosphere where frequencies near the line center have already been strongly attenuated)? It would be helpful to clarify this point.

8. Page 12, Line 20: Recommend changing "...signal significantly above the SNR" to "...signal significantly above the noise floor".

   Same line: Recommend changing "...reduced value of the tones further away..." to "reduced absorption for tones further away..."

9. Page 12, Line 31: To say that for fixed duty-cycle there is improvement in going from two to four frequencies is misleading. In the 2-frequency case, one cannot perform the 3-parameter fit, and therefore the frequency-dependent hydrometeor scattering effects enter directly as a bias in the retrieved humidity. Thus, these two situations are fundamentally different.
* * *

---

## Referee Comment (RC2) · Anonymous Referee #2 · 26 Mar 2019

This paper develops an idea from previous studies to evaluate the potential benefit of a differential absorption radar system near 183GHz to measure water vapour in ice-phase clouds. This is an important application, and the technology and idea are new and emerging so it is timely to do this kind of study.

What this paper does well is delivery of the "headline" results that: retrieval of humidity in ice cloud is possible; that the precision of RHice is at a level where it is microphysically useful; that looking down from space gives quite different sensitivies and error characteristics than looking up from the ground.

What the paper does not do so well are:

[Figure]

(i) placing this work in the context of what has been done before. Previous papers are reviewed briefly, but what is not so clear is the exact point of departure of this paper vs the previous studies. In particular I would like to see the current paper clearly distinguished from the previous papers by Roy et al and Millan et al, because these both consider water vapour retrieval in the presence of cloud and precipitation, and the Millan study includes ice which is also your focus. It needs to be clear (a) what aspects of the retrieval methodology are new, and (b) what aspects of the way the performance of the retrieval are analysed are new

(ii) clearly explaining the sensitivity of the results to assumptions made, or the generality of the conclusions drawn. For example a particular ice particle scattering model is assumed, and some linear fit between dB of attenuation (per unit IWC, per unit path length) vs frequency is made A+Bf. What is not explained are - the justification for that scattering model; the change in the results if you'd made a different choice; whether the numerical values of A and B matter, or just the fact that the behaviour is linear.

I would like to see this paper published, but I would also like to see it revised significantly and expanded to address some of these points.

Some specific comments:

In the abstract you mention an airborne system in the same breath as a spaceborne system, and in the paper you concentrate on the satellite option. But these two setups are actually quite dissimilar. A satellite moves at about 7km/s relative to the earth's surface, which means even relatively few pulses lead to quite large pixels. A research aircraft flies at speeds of about 100m/s - seventy times slower. So in principle you could do a lot more averaging of pulses, which improves your number of independent samples. It would be interesting to consider a separate "airborne" scenario in addition to your ground and space views.

Page 2 line 25. "...coupled with that of temperature" - I agree with the other reviewer that the impact of not knowing T perfectly (e.g. estimating from reanalysis/forecast

model with 1 or 2K typical uncertainty) needs to be discussed somewhere.

Page 2 line 28. You start talking about supercooled LWC here. Is this factor included in the retrieval? or does that 0.5dB differential not matter relative to the size of signal you are estimating?

Page 3 line 2 "could help us understand how the ice crystal grow significantly enhance water mass fluxes due to sedimentation" - this sentence needs rewording

Page 3 line 5, and figure 1. You have constructed some sort of Magono-Lee style diagram in the figure here, but this is not a particularly accurate representation of our current state of knowledge. Specifically, at temperatures below -20C or so the crystals are almost always polycrystalline, which may be in the form of multiple plates, or bullet rosettes (the column polycrystal form). This was recognised a long time ago (e.g. Aufm Kampe et al 1951 J. Meteorol.) and has been reiterated by e.g. Bailey and Hallett (2009 JAS). Minor points - one of your images, which I suspect is meant to show a column, actually shows a capped column. This happens when a column is transported (by convection, or sedimentation) to a temperature favouring plate growth. This touches on a problem for your idea of using T,RHice to constrain the crystal types in the cloud - in a deep ice cloud like your case study in figure 3, particles are growing and falling through temperature changes of 10s of K across their lifetime. The habit diagram in figure 1 is for isothermal growth in the lab. Connecting the two is not simple. Indeed in your retrieval you actually assume the particles are aggregates... which do not even appear on your habit diagram...

Figure 1: A simple, but useful exercise to add here, would be to give some indication of how the growth rates of the crystals depend on RHice and T. You could show indicative calculations for a few temperatures and crystal size of a few hundred microns. Then you could perturb the calculation by your expected uncertainties (q +/- 3%) and see how the growth rate is affected. Ultimately the uncertainties will be most important for RHice close to 100%, and the more sub- or super-saturated you are, the less significant

they will become.

Page 4 line 14. Multiple scattering neglected because footprint is small and SSA is low. Can you foresee any scenario where these would become significant? I'm surprised CloudSat suffers from multiple scattering at 94GHz, yet this G-band system would not, especially if there is a substantial optical depth to the cloud

Page 5 line 7 and Figure 2. This calculation needs explanation. What are we assuming here and why? Various Dm values are shown. What size distribution is assumed here? The caption tells us this is from Leinonen and Szyrmer's study, but with "LWC=0" - so I infer no riming. It would be better to explain these things directly. Then later on, you say you are using SSRGA for the ice scattering. This seems to be a contradiction? Are you somehow using both? Or SSRGA tuned to L&S's aggregates?

Figure 2: I was quite surprised both at the amount of attenuation here, and its frequency dependence. What physical effect is driving this - absorption or scattering? Is there any sensitivity of this to the scattering model assumed?

Figure 4 - show ground based simulated Z's here as well, to match figure 6

Page 11. It could be useful to explore the trade off between long vs short pulses. Long pulses = better SNR which seems to be a critical parameter. But shorter pulses = more range gates = better fit to attenuation profile (and implicitly more independent pulses in total)?

Page 12 ground based system assumed to be at 270K level. Why? Are the data from everywhere in globe, or a particular latitude band? I would expect the performance trade offs you touch on to change a lot between the tropics and polar regions

The discussion of figure 8 is very brief. Please expand and carefully explain what you want the reader to take from this, and how the plot supports that argument

Minor points

I'm not a spectroscopy person, but the use of "left" and "right" in relation to the wings of the absorption line seemed odd. I guess you are referring to higher and lower frequencies? or is the diagram in your head in wavelength (in which case, it's the opposite way around!)

Page 9 line 8 tidy up brackets ] -> [

Figure 5. When I printed this half the figure disappeared. I'm not sure if this reflects an underlying issue with the figure file. I could view it OK on the screen however. Note that in the lower panel, the legend for the 94GHz profile is wrong (cyan line rather than black x)

Figure 8 caption - dashed region -> shaded region

[Figure]

---

## Author Comment (AC1) · 9 May 2019

Our reply is in the attachment together with the revised version of the paper.

Please also note the supplement to this comment:
https://www.atmos-meas-tech-discuss.net/amt-2019-14/amt-2019-14-AC1-supplement.pdf

---

## Author Response (AR1)

**Responses to reviewer 1**

Evaluation of differential absorption radars in the 183 GHz band for profiling water vapour in ice clouds

A. Battaglia, P Kollias

May 9th, 2019

We thank the reviewer for his/her interesting and insightful comments.

Hereafter a point to point response (reviewers' comment in black, response in red). We also attached a revised version of the paper with changes in bold.

**Comments from reviewer 1**

1. As a general comment for G-band radars, it is important to acknowledge the international frequency allocation restrictions that prohibit transmission within certain frequency ranges. Since this has impacted similar technologies in recent years, it seems reasonable to address this issue in the introduction to the paper.

   Yes we are aware about this issue: it was indeed reported in the conclusions. It has now been mentioned in the introduction as well (See page 3, line 9-11).

2. Page 4, Line 1: The discussion of relative humidity measurement precision only addresses the role of absolute humidity measurement uncertainty. However, the temperature uncertainty is likely to dominate the error in RH. For instance, assuming the absolute humidity (i.e. water vapor density) is known perfectly, an error of 1 K in the temperature would lead to an error of 8% in RH at 260 K. This is much larger than the 3% level that is used as a metric for good accuracy in the paper. How would coincident temperature measurements be performed for precise RH studies? Since a principal goal of the paper is to show the utility of DAR for retrieving RH (and not simply water vapor density) for ice microphysics studies, this point needs to be addressed.

   Thanks for pointing this out. We have relaxed our requirements for RH to 5-10% and we have discussed the uncertainties introduced by temperature on the RH retrieval (see new text at page 4, lines 7-19).

3. Page 5, Line 9: The authors use the phrase "mean square fitting procedure" a couple times in the text. Do they mean to say "least squares fitting procedure"?
   Yes sorry for the confusion, corrected.

4. Page 5, Line 11: What sets the necessary frequency span to be 10 GHz if one wants to allow for the linear term with coefficient B? Shouldn't this depend on where the window of frequencies is positioned relative to the line center?
   The term B is accounting mainly for the frequency dependence of the scattering properties of the hydrometeors and of dry air. As a result we do not expect that this is very much affected by the proximity to the water vapour line. Scattering properties of hydrometeors are slowly varying with frequency as shown in Fig.2, therefore this is not related to how close we are to the absorption line.

5. Page 7, Line 1: The issue of sacrificing duty-cycle for an increased number of frequencies in DAR is very important. It is unclear what the authors mean when saying that sensitivity can be

held constant while increasing the number of frequencies by "increasing the duty cycle of the radar." Presumably, one wouldn't want to sacrifice the current range resolution of roughly 500 m, which means the pulse width cannot be lengthened from 3.3 µs. Additionally, since the pulse time of-flight through 10 km of atmosphere takes roughly 70 µs (2-way), how can the PRF be increased much from 6 kHz? Furthermore, the implementation of frequency diversity is technologically nontrivial for the large number (up to 7) and range (≈ 15 GHz) of frequencies proposed in this work. Since the large frequency range is critical for the 3-parameter fitting routine, and since a reduction in the sensitivity per channel by a factor of √ 7 would certainly affect the DAR measurement precision, a comment on the technical feasibility of such a system is needed.

We agree with the reviewer comment about the confusing statement regarding the use of frequency diversity and increasing the radar duty cycle. A more detail statement regarding some important technical aspects of the spaceborne G-band radar has been added in the manuscript (see new text at page 9, lines 11-21).

"The DAR system shown in Table 1 is for a 2% duty cycle, 100 W peak output power Extended Interaction Klystron (EIK) system. The Communications and Power Industries (CPI) has about 15 GHz bandwidth. Selecting a number of tones (i.e. 4) is technologically feasible using a single chirp generator and four intermediate chains which are switched between to select the tone. The switching can be done from pulse to pulse.

In addition to high power sources (EIK), lower power sources are available either using frequency multipliers coupled with commercially available amplifiers or microwave sources/oscillators (Virginal Diodes, https://www.vadiodes.com/en/). Recently, Jet Propulsion Laboratory developed a new power approach using GaAs Schottky diode frequency 2x multipliers at 183 GHz (Cooper et al., 2016).

Preliminary estimates of the expected radar sensitivity using these different architectures indicate that a minimum sensitivity of -22 dBZ is possible for 4 different tones. The impact of reducing sensitivity to accommodate additional tones is discussed in section 4.''

Cooper K. B., S. Durden, M. Choukroun, M. Lebsock, J. Siles, R. Monje, and C. Lee, 2016: FMCW Radars at 95 and 183 GHz for Planetary and Earth Science Remote Sensing. 2016 Global Symposium on Millimeter Waves (GSMM) & ESA Workshop on Millimetre-Wave Technology and Applications."

6. Page 9, Line 15: The statement that water vapor can be retrieved to better than 15% accuracy in regions of the atmosphere where the water vapor density varies by more than one order of magnitude can be misleading. An important property of the DAR measurement method is that the measurement parameters (specifically the transmit frequency locations and number of pulses per frequency) and the local pressure and temperature determine the maximum achievable absolute precision in water vapor density for a given spatial resolution (i.e. value of Δr). This has the consequence that for a given measuring system, lower values of absolute humidity will be measured with lower relative precision. The important relationship between absolute humidity value and relative precision of the DAR measurement should be discussed. As a related point, while the line-fitting retrieval routine implemented in this work gives an estimate of the measurement precision, this information is not discussed. The authors do go into great detail about the measurement biases (i.e. accuracy) in the form of relative error plots, but do not present results on the measurement precision.

It would be useful to include a corresponding plot of measurement precision in Fig. 6 along with the relative error plot. Do the statistical errors from the line-fitting procedure agree well

with the scatter of measured values relative to "truth." Such a discussion would elucidate the difference between systematic biases and random error in the measurement.

We have now clarified some confusion in our first version (where we erroneously used the word accuracy a couple of times). In our simulation frameworks we basically assess how precise our estimates are, given the expected noise level of the measurements. Previous work by Millan et al. has indeed demonstrated that potential biases are expected to be smaller than the precision of the technique (see their Fig. 7), and therefore precision is the major roadblock for this technique: in other terms, can we beat the noise of the measurements to the level where we can get something useful out of the system?

The accuracy of a given retrieval can indeed be evaluated only via comparison with real measurements (which is not the case in a notional study like the one here proposed). Some of the potential biases can be assessed (e.g. by looking at the impact of using different absorption models, but it is not within the scope of the current paper). Precision on the other hand come straight from the fitting routine based on the noise level of the measurements. This has been clarified in the text (see page 4, lines 20-27). In fact this generalises the computation of the precision when only two tones are available (now formula 7). It is true that if a fixed pair is considered then the relative precision is becoming increasingly lower with lower values of water vapor density but when considering profiling capability different channels will be involved: retrieval values of smaller vapor densities will give more weight to channels close to the centre of the line and therefore to larger values of extinction coefficient. The trick when water vapor profiling for a broad range of water vapor densities is indeed in finding the right balance between signal and strength of the absorption.

7. Page 12, Line 19: It is confusing to read that results worsen for high reflectivities, where one usually expects the SNR to be large. Maybe what is meant here is that regions of high reflectivity are also associated with high absorption (or are lower in the atmosphere where frequencies near the line center have already been strongly attenuated)? It would be helpful to clarify this point.

Yes the confusion arises from the fact that the reflectivities we are referring to are ``CloudSat reflectivities''. Of course that region is typically located in the lower troposphere where there is strong absorption due to water vapour. The sentence has been amended, see text at page 15, line 10-13.

8. Page 12, Line 20: Recommend changing "...signal significantly above the SNR" to "...signal significantly above the noise floor". Same line: Recommend changing "...reduced value of the tones further away..." to "reduced absorption for tones further away..."

Corrected.

9. Page 12, Line 31: To say that for fixed duty-cycle there is improvement in going from two to four frequencies is misleading. In the 2-frequency case, one cannot perform the 3-parameter fit, and therefore the frequency-dependent hydrometeor scattering effects enter directly as a bias in the retrieved humidity. Thus, these two situations are fundamentally different.

This has been now clarified in the text, see page 15, lines 28-33.

**Responses to reviewer 2**

Evaluation of differential absorption radars in the 183 GHz band for profiling water vapour in ice clouds

A. Battaglia, P Kollias

May 9th, 2019

We thank the reviewer for his/her interesting and insightful comments.

Hereafter a point to point response (reviewers' comment in black, response in red). We also attached a revised version of the paper with changes in bold.

**Comments from reviewer 2**

**General comments**

Thanks for this general comments. We have now expanded the introduction trying to make clear what is the goal and the paper and where it is different from previous research (see new text at page 4, lines 20-27).

**Specific comments**

1. In the abstract you mention an airborne system in the same breath as a spaceborne system, and in the paper you concentrate on the satellite option. But these two setups are actually quite dissimilar. A satellite moves at about 7km/s relative to the earth's surface, which means even relatively few pulses lead to quite large pixels. A research aircraft flies at speeds of about 100m/s - seventy times slower. So in principle you could do a lot more averaging of pulses, which improves your number of independent samples. It would be interesting to consider a separate "airborne" scenario in addition to your ground and space views.

   We have introduced a section in the discussion (Sect3.1, page 13) where we refer also to the possibility of adopting an airborne configuration. The review is absolutely correct: the key advantage for an airborne scenario is indeed to collect more pulses and therefore to significantly improve the precision of the measurements (or otherwise to improve the resolution of the measurements).

[Figure]

2. Page 2 line 25. "...coupled with that of temperature" - I agree with the other reviewer that the impact of not knowing T perfectly (e.g. estimating from reanalysis/forecast model with 1 or 2K typical uncertainty) needs to be discussed somewhere.
This has been discussed now at page 4, line 5-19.

3. Page 2 line 28. You start talking about supercooled LWC here. Is this factor included in the retrieval? or does that 0.5dB differential not matter relative to the size of signal you are estimating?
Supercooled LWC were not included in the microphysical scenario but the eventual presence of a SLWC layer will indeed be captured by the retrieval via the B-term in Eq.6. A typical supercooled layer with 100 g/m^2 LWP will indeed produce a difference between two channels within 10 GHz of less than 0.1 dB.

4. Page 3 line 2 "could help us understand how the ice crystal grow significantly enhance water mass fluxes due to sedimentation" - this sentence needs rewording
Sentence has been rephrased.

5. Page 3 line 5, and figure 1. You have constructed some sort of Magono-Lee style diagram in the figure here, but this is not a particularly accurate representation of our current state of knowledge. Specifically, at temperatures below -20C or so the crystals are almost always polycrystalline, which may be in the form of multiple plates, or bullet rosettes (the column polycrystal form). This was recognised a long time ago (e.g. Aufm Kampe et al 1951 J. Meteorol.) and has been reiterated by e.g. Bailey and Hallett (2009 JAS). Minor points - one of your images, which I suspect is meant to show a column, actually shows a capped column. This happens when a column is transported (by convection, or sedimentation) to a temperature favouring plate growth. This touches on a problem for your idea of using T,RHice to constrain the crystal types in the cloud in a deep ice cloud like your case study in figure 3, particles are

growing and falling through temperature changes of 10s of K across their lifetime. The habit diagram in figure 1 is for isothermal growth in the lab. Connecting the two is not simple. Indeed in your retrieval you actually assume the particles are aggregates... which do not even appear on your habit diagram...

Criticism accepted: we have got rid of all the shapes in Fig1 and modified the text at page 3 along the reviewer's suggestions. See new text at page 3 , third item in the list.

6. Figure 1: A simple, but useful exercise to add here, would be to give some indication of how the growth rates of the crystals depend on RHice and T. You could show indicative calculations for a few temperatures and crystal size of a few hundred microns. Then you could perturb the calculation by your expected uncertainties (q +/- 3%) and see how the growth rate is affected. Ultimately the uncertainties will be most important for RHice close to 100%, and the more sub- or super-saturated you are, the less significant they will become.

Growth by the vapour deposition is described by the differential equation (e.g. see Field et al., JAS 2008):

$$\frac{dm}{dt} = 4\pi C \; S_i f\left(Re, T\right)$$

where C is the capacitance of the ice crystal, f is a function of the Reynolds number and the temperature, and $S_i$ is the supersaturation with respect to ice. As a result dm/dt is linearly proportional to $S_i$. A +-3% change in $S_i$ will therefore induce a +-3% change in the growth rate. In terms of size of the ice crystals we have looked at the increase/reduction of an ice crystal of 500 micron size in different conditions of $RH_i$. Results are plotted in the new right panel of Fig.1.

7. Page 4 line 14. Multiple scattering neglected because footprint is small and SSA is low. Can you foresee any scenario where these would become significant? I'm surprised CloudSat suffers from multiple scattering at 94GHz, yet this G-band system would not, especially if there is a substantial optical depth to the cloud.

Multiple scattering effects are driven not by the optical depth of the cloud but by its scattering optical depth. Since most of the optical depth is indeed due to absorption it does not generate multiple scattering. For instance in the profile shown in Fig. 5 we do expect multiple scattering to occur more likely for the channels away from the line centre. The footprint of our space-borne configuration is of the order of 400m, so significantly less than for CloudSat, again suppressing multiple scattering contributions. We do expect such effects to really occur in the region of high CloudSat reflectivities (above 10 dBZ) where the water vapor profiling is indeed expected not to perform greatly anyhow. For CloudSat, multiple scattering in ice clouds is marginal (see Matrosov and Battaglia). We have added also a suggestion how to identify multiple scattering. New text has been introduced at page 5, lines 8-13 to explain this.

8. Page 5 line 7 and Figure 2. This calculation needs explanation. What are we assuming here and why? Various Dm values are shown. What size distribution is assumed here? The caption tells us this is from Leinonen and Szyrmer's study, but with "LWC=0" – so I infer no riming. It would be better to explain these things directly. Then later on, you say you are using SSRGA for the ice scattering. This seems to be a contradiction? Are you somehow using both? Or SSRGA tuned to L&S's aggregates?

The assumed size distribution for ice crystals is an exponential particle size distribution, this has been clarified in the caption. An additional scattering model (more useful for aggregates) has been added to show the strong dependence of the scattering model of the ice extinction properties. The supercooled attenuation coefficient has also been revised (the Ellison 07 model has been used for the refractive index of water: this provides a better agreement with radiometric measurements as demonstrated in Turner et al, 2016). For the single scattering properties we are using SSRGA for different particle models as generated by Leinonen and Szyrmer. Their particle model is used to compute the SSRGA parameters that describe the

distribution of mass along the radiation incident direction. See new figure 2 plus text at page 8, lines 3-7.

9. Figure 2: I was quite surprised both at the amount of attenuation here, and its frequency dependence. What physical effect is driving this - absorption or scattering? Is there any sensitivity of this to the scattering model assumed?
The attenuation is mainly caused by scattering and yes it is strongly scattering model dependent (a different scattering model has now been shown in Fig.2). new text at page6, lines 6-10.

10. Figure 4 - show ground based simulated Z's here as well, to match figure 6
Two panels have been added to Fig4 as requested. Text has been added at page 10, lines 1-4.

11. Page 11. It could be useful to explore the trade off between long vs short pulses. Long pulses = better SNR which seems to be a critical parameter. But shorter pulses = more range gates = better fit to attenuation profile (and implicitly more independent pulses in total)?

The trade-off has been explored (compare bottom panels in Fig.6 where a configuration with a 120m and a 480 m pulses is considered.). Clearly it is much better to use pulses that match the required vertical resolution because SNR is a critical parameter for the precision of the measurement. Averaging more gates do not recover the same precision. See new text at page13, lines 28-32

12. Page 12 ground based system assumed to be at 270K level. Why? Are the data from everywhere in globe, or a particular latitude band? I would expect the performance trade offs you touch on to change a lot between the tropics and polar regions The discussion of figure 8 is very brief. Please expand and carefully explain what you want the reader to take from this, and how the plot supports that argument.
The selection of the 270 K level was totally arbitrary. Also data are taken from everywhere in the globe (above such temperature level). The idea was to assess the potential of a DAR system when operated from a ground-based at freezing temperature. Of course the selection of the tones could be optimized for a specific location and time of the year based on the local cloud and temperature climatology. This aspect has now been noticed at page 17, lines 9-11.

Minor points

I'm not a spectroscopy person, but the use of "left" and "right" in relation to the wings of the absorption line seemed odd. I guess you are referring to higher and lower frequencies? or is the diagram in your head in wavelength (in which case, it's the opposite way around!)
We change the term ``right'' to ``upper'', it is always in junction with ``183 GHz'' band so it should be implicit that we are thinking in the frequency domain.

Page 9 line 8 tidy up brackets ] -> [

Done

Figure 5. When I printed this half the figure disappeared. I'm not sure if this reflects an underlying issue with the figure file. I could view it OK on the screen however. Note that in the lower panel, the legend for the 94GHz profile is wrong (cyan line rather than black x)

Done

Figure 8 caption - dashed region -> shaded region

Corrected

[revised manuscript text omitted]

---

## Author Response (AR2)

**Responses to reviewer 3**

**Evaluation of differential absorption radars in the 183 GHz band for profiling water vapour in ice clouds**

A. Battaglia, P Kollias

May 9th, 2019

We thank the reviewer for his/her final minor comments.

Page 1 Line 19. …before significant progress *can* occur…

DONE

2. Figure 1 caption. Should this caption be changed to reflect the fact that the Magono and Lee (1966) ice crystal small photographs are no longer shown in this figure?

Caption to the figure has been changed

3. Page 4 Line 2. Change "tenths" to "tens"?

DONE

4. Page 4 Line 30. Change "reflectivities profiles" to "reflectivity profiles"

DONE

5. Page 6 Lines 30-31. Slightly reword to economize and improve grammar: Our approach uses ice microphysical properties from spaceborne sensors as input to a forward radar model (DAR model) to generate reflectivities around the 183.3 GHz absorption band.

Sentence has been reworded

6. Page 15 Line 21. Wording suggestion: replace "is going down" with "decreases". Also fix "becasue" typographical error

DONE